# Cascading Recycling of Wood Waste: A Review

**DOI:** 10.3390/polym13111752

**Published:** 2021-05-27

**Authors:** Arnaud Besserer, Sarah Troilo, Pierre Girods, Yann Rogaume, Nicolas Brosse

**Affiliations:** 1LERMAB, Université de Lorraine, INRAE, GP4W, F 54 000 Nancy, France; arnaud.besserer@univ-lorraine.fr (A.B.); sarah.troilo@cf2p.eu (S.T.); 2LERMAB, Université de Lorraine, INRAE, ERBE, F 88 000 Epinal, France; pierre.girods@univ-lorraine.fr (P.G.); yann.rogaume@univ-lorraine.fr (Y.R.)

**Keywords:** wood waste, urea-formaldehyde resin, cascade effect, medium density fiberboards, recycling

## Abstract

Wood is an increasingly demanded renewable resource and an important raw material for construction and materials. In addition, new consumption habits are leading to the production of ever greater volumes of waste wood, which constitutes a feedstock that can be mobilized for the cascade production of new materials such as particleboard. However, current legislation and wood waste recycling processes need to be improved in order to maximize the volumes that can be reused and to upgrade the properties of the recycled wood. This review describes wood waste flows and volumes available in Europe, the current French and European legislation, and the innovations under development in this field: innovative automated sorting techniques, physical-chemical processes for cleaning residual glue from the surface of wood particles, cleaning of MDF, and bioremediation processes for cleaning hazardous wood contaminated by heavy metals or creosote.

## 1. Introduction

Wood has many advantages in relation to the concepts of the bio-economy and circular economy. It is a material of natural and renewable origin, biodegradable, with remarkable mechanical and thermal characteristics. The environmental impacts during the production and end-of-life phases of the wood material are generally much lower than those of equivalent materials produced from inorganic or fossil raw materials [1]. Moreover, unlike resources of agricultural origin, wood does not compete with food. As a result, since the beginning of the 21st century and in addition to traditional uses, there has been an increase in the consumption of wood for new applications (energy production, building materials, chemicals, etc.) [2]. A study shows that by 2030, the production of wood could be insufficient to meet demand in Europe [1]. The increase in wood consumption is accompanied by an increase in the production of waste wood from end-of-life wood-based products. Recycling this large deposit could thus constitute a source of abundant and inexpensive raw material for the production of new materials.

Although wood is a natural material, waste wood generally contains additives (glues, varnishes, and paints), various pollutants (wood treatment products and heavy metals), and contaminating materials (glass, plastics, metals, etc.). This heterogeneity greatly complicates recycling processes. Consequently, current wood waste management strategies are mainly based on (1) landfill, (2) energy recovery, and (3) material recovery [3]. Even if it is more complex, the latter route of recovery is to be developed because it is based on recycling through the production of new materials and consists of a “cascade” use. According to [4] cascading use is “the efficient utilization of resources using residues and recycled materials for material use to extend total biomass availability within a given system”. As a result, the cascade effect makes it possible to prolong the storage of carbon in the material, thus delaying its release in the form of CO_2_ during its end-of-life energy recovery [5] (Figure 1).

The primary reuse of recycled wood is currently in the particleboard industry. As an example, a recent paper reports the incorporation of construction and demolition wood waste in the inner layer of medium density fiberboard [6]. In Europe, the consumption of particleboard in 2019 was 37.07 million m^3^ [7]. The proportions of recycled wood in particleboard are very diverse depending on the country. It varies from ≈100% in Italy; ≈50% in Belgium, the United Kingdom, and Denmark; 15–30% in Germany, France, and Spain; and 0% in Switzerland [4]. It appears that there is room for improvement in many European countries. Other developments, much smaller in volume, regarding cascading use of waste wood concern the production of waste wood-plastic [8,9,10,11] or waste wood-concrete composites [12].

This review article therefore focuses on the recycling of wood waste for the production of new materials. It describes the current situation in Europe in terms of collection and reuse flows and regulations. This article also describes ongoing research and innovation aimed at improving existing processes and expanding the range of recyclable wood waste to reduce volumes destined to landfill and combustion. Advanced methods under development for automatic spectroscopic sorting, particle, and fiber cleaning using physical-chemical, thermal, and biological techniques are described.

## 2. Waste Wood Resource

Wood wastes are divided into two main categories: industrial wastes generated within the industry and final wastes, after use of the products. The majority of the first are considered as by-products and are not legally defined as waste; they will not be considered in this article. As a reminder, a waste is defined as “any substance or object which the holder discards or intends or is required to discard” [13]. This same directive defines a hierarchy of waste prevention and treatment methods in order of priority: reduction in production and toxicity, reuse, recycling, other recovery, including energy, and finally elimination.

At European level [14], the amount of wood waste was estimated to be about 33.2 million tons in 2007, with significant disparities between countries: about 55–60 kg/inhabitant/year in Eastern and Southern countries, respectively, up to 75 kg/inhab/year for Western countries, and 110 kg/inhab/year in Northern countries. On a European scale, the treatment methods are in the following order: (1) disposal (landfill and incineration) for 37%, (2) material recovery (mainly particle boards) for 33%, and (3) energy recovery (heat production or cogeneration) for 30%. In Eastern and Southern countries and in the United Kingdom, waste is mostly buried, whereas in Northern and Western countries, recovery is more important, in terms of materials in Italy and France and in terms of energy in Germany, Sweden, and Finland. Figure 2 shows the quantities and treatment methods in the 28 EU countries in 2010 [4].

A better knowledge of wood waste composition and quality is critical. The impurities and contaminants considerably vary with wood waste origin. Wood waste should not be considered as a homogeneous material, but rather be managed as a complex and variable material flow [15]. The construction/demolition sector is the largest contributor of wood waste. In fact, in Europe, wood fraction represents 20–40% of construction/demolition waste. Furniture industry is also an important contributor of waste. A smaller proportion comes from packaging [15]). Table 1 gives a classification of wood wastes according to their origin.

Wood recycling very generally requires a reduction in size to small particles (chips, fibers, etc.) that can be reused for the production of composite materials. The specific topic of size reduction in wood waste was recently review ([16]).

## 3. Waste Wood Legislation in Europe

One of the main challenges to optimize the recovery of wood waste is related to its classification, which is not harmonized at the European level. Generally speaking, non-hazardous and hazardous waste are separated into two very different classes, and then, in between, more or less contaminated waste represents one or two classes depending on the country. These classes are mainly related to the uses and thus to the regulatory limits for the recovery in panels or energy most often. For example, in France, the usual classification shows 3 classes A, B, and C, defined as follows:

Class A: clean products (without additives)

Class B: lightly admixed products

Class C: heavily admixed products.

Class A refers to clean or very lightly treated wood such as pallets containing chipboard blocks, e.g., these are mainly packaging materials such as crates, pallets, etc. At the other end of the spectrum, class C designates potentially dangerous woods that contain substances classified as dangerous: these are mainly woods that contain heavy metals (e.g., CCA copper-chromium-arsenic treatments) or creosote. Class B is thus defined by difference as containing all the waste, which is neither wood A nor wood C. This class is thus very broad and designates all wood containing additives such as glues, paints, finishing products, veneers, etc. Therefore, in this class of wood, very little polluted wood that contain only a few traces of paints or varnishes as well as wood that is heavily contaminated by glues, coatings, or even PVC edges is present.

To try to overcome this heterogeneity, most of the studies or projects in progress use the regulations, standards, or thresholds of the most common recovery routes, i.e., recycling into panels or energy recovery. For particleboard, the European Panel Federation [17,18] imposes on its members to respect a specification with high limits of presence of some compounds in recycled wood, as well as in the panels. These limits are given in Table 2 and are derived from the regulations related to the use of toys by children with mouth contact [19] and the EN 71-3 standard (2019) [20]. These limitations mainly concern metals, halogens, pentachlorophenol (PCP), and polycyclic aromatic hydrocarbon (PAH).

For the energy recovery, the regulation exists for each country and is defined in the environmental regulation code. For example, in France, the ICPE (Installations Classified for the Environmental Protection) defines several headings according to the quality and the nature of the fuel. Wood waste can thus correspond to 4 different headings:-Heading 2910-A: this is clean wood from forest resources or industrial by-products, or even waste that has undergone an SSD (Exit from Waste Status) process. It corresponds globally to the wood waste of class A;-Heading 2910-B: this involves the use of non-hazardous and very slightly contaminated waste, generally within the companies themselves. The fuels do not contain any organo-halogen elements and no or only traces of metals. These installations thus correspond to very slightly contaminated B-wood, with limits to be respected, in particular, as regards the contents of metals, chlorine, PCP, and PCB;-Heading 2971: this heading is recent (2016) and concerns what is called RDF (Refused Derived Fuels). These same RDF are regulated according to the EN 15,359 standard [21], which proposes 5 classes according to their calorific value and their chlorine and mercury content. It should also be noted that the sale of RDF must be accompanied by a form on which other elements are requested, such as the origin, the ash content, the metal, nitrogen and sulfur content, etc. These installations are thus usable for wood B contaminated in variable proportions but not by dangerous products;-Heading 2770: this heading concerns incineration, the wood wastes concerned being hazardous wastes, therefore of class C.

Today, many works are in progress to improve the classification of wood waste in order to facilitate its recovery in the different sectors while respecting both the hierarchy of uses and the protection of health and the environment. The general idea is to establish 4 classes based on the previous quality elements (recycling or energy) and on what exists in other European countries such as Finland, Germany, or the United Kingdom. Table 3 from [22], is the one we are moving towards.

The implementation of a more relevant classification on the scale of the European continent should make it possible to promote the recovery of this waste. The development of sorting and recovery technologies is also a strong opportunity to support the development of this sector.

## 4. Hydrolysis and Thermolysis of Residual Adhesives

The main contaminant of recycled wood is the glue used in particleboard, plywood, fiberboard, etc. Urea-formaldehyde (UF) glue used alone or reinforced with small quantities of melamine is the most commonly used. The main advantages of UF glue are its low cost, absence of color, and high chemical reactivity during polymerization. The more expensive phenol-formaldehyde or pMDI-based adhesives are used for specific applications, particularly in wet environments. In fact, the main disadvantage of UF adhesives is their low resistance to chemical hydrolysis, which limits their use in outdoor applications. In the presence of moisture, the UF polymer chain can undergo a hydrolysis reaction described in Figure 3, leading to shorter polymer fragments and a drop in the mechanical properties of the wood-UF composite.

The presence of residual glue on the surface of the wood fibers or particles makes their recycling more difficult, especially for the production of new wood panels. The cross-linked glue particles do not react with a new adhesive to form a polymer network during a second cross-linking step, resulting in a significant drop in the properties of the materials formed from recycled panels. This negative impact of the residual UF on the fibers surface has been confirmed by gel time measurements [24]. It has been shown that the presence of only 6% by weight of residual UF glue results in a reduction of up to 50% in the shear strength of a plywood panel [25].

The hydrolysis susceptibility mentioned above can then be used to dispose of the UF adhesive in a recycling process. A removal of two-third of the UF adhesive by simple treatment in water at a temperature of <100 °C has been described [23,26]. Lubis et al. studied the hydrolysis of cured resins and showed that the pH of the medium strongly influences the hydrolysis kinetics according to the following order: acid > neutral > alkali. The optimum conditions of their study to remove UF resin from MDF were: 80 °C for 2 h in the presence of oxalic acid [27]. This hydrolysis reaction can be accelerated by treatment at higher temperatures using water vapor. For example, treatment in the presence of steam at temperatures of 150–190 °C for 10–20 min will remove 80% of the adhesive [28]. The main limitation of this approach is related to the concomitant degradation of the wood fiber (in particular, hydrolysis of hemicelluloses), which affects the morphology (particle or fiber size) and properties of the wood.

Low-temperature pyrolysis could also be considered for partial depollution of wood waste [29,30,31]. Pyrolysis is a degradation reaction under the effect of heat and in an inert atmosphere, i.e., without oxygen. It is often carried out under nitrogen (N_2_), argon, and helium being also commonly used.

Figure 4 shows the degradation rate of wood, UF resin, and melamine formaldehyde (MF) resin during the pyrolysis reaction performed in thermogravimetric analysis (TGA) device. It is clearly visible that the UF resin begins to degrade at a lower temperature than wood. When the pyrolysis takes place between 250 and 300 °C (gray zone), the degradation of the resin is then promoted [32,33,34]. This method allows up to 60% of the resin to be removed while limiting the degradation of the wood for treatment times between 8 and 15 min and temperatures between 250 and 300 °C [32]. The main pyrolysis products of wood-based panels pyrolysis, for gaseous form, are NH_3_, HCN, and HNCO [29,30] particularly, NH_3_ has significantly higher yields than HCN, especially under lower pyrolysis temperatures. During the degradation process, initial UF polymer chains react with lignin to produce heterocyclic-N [33,35], thus intimately bonding the residual resin to the wood and making it unavailable to be removed by hydrolysis. Due to the higher temperature levels in pyrolysis process, the wood is more degraded by this method than by hydrolysis.

Considering MF resin, the temperature range of degradation is over that of the wood (i.e., Figure 4), therefore, the pyrolysis process is inefficient for MF resin curing.

## 5. Wood Fibers and MDF

An effective process to homogenize wood waste is defibration. Fibers are good candidates for the development of new recycling routes [36,37,38,39] as they are the raw material for many products, such as paper, MDF (medium-density fiberboards), and wood fiber-based insulation boards (commonly called “wood wool” in reference to their glass or rock wool equivalent), already on the market. Other applications of wood fibers also exist such as the reinforcement of composites (plastic, concrete, etc.) or the production of textiles for clothing (Tencel© and Refibra©). However, there are two major concerns with recycling wood through fiber production. (1) The lengths of fibers obtained from waste wood are often shorter than those obtained from native wood. (2) The fibers produced retain debris such as plastics or decorative elements (e.g., melamine coating). They also still contain some of the additives originally contained in the waste wood (glues, paints, fungicides, etc.).

These limitations make difficult the integration of wood fibers produced from wood waste into the industrial manufacturing cycle. On a small scale, the technical feasibility of producing wood fiber-based insulation panels, wood/plastic composites, and MDF and Kraft pulp from furniture waste has been demonstrated in research projects. However, the economic viability of these different recovery routes needs to be demonstrated.

The presence of MDF in the mixture of waste wood poses a problem for recycling in the form of chipboard. With the evolution of technology, MDF sorting is now possible on an industrial scale, which will eventually allow the panel industry to increase the rate of waste integration and reduce the constraints on the MDF content of incoming waste, which is currently set at 3%. Fiber production from waste MDF is easier and less energy intensive than the traditional mechanical wood defibration process. Treatments of MDF by steaming or hydrolysis followed by mechanical defibration (refiner and hammer mill) have been described for the production of recycled MDF boards. A recycled fiber integration rate > 20% without a significant decrease in the mechanical properties of the MDF produced has been described [36]. Recently, Hong et al. successfully produced three-layer MDF panels from hammer-milled surface-laminated MDF. The authors showed that the substitution of >20% recycled fiber results in panels with improved properties in terms of formaldehyde emission and thickness swelling [40]. MDF Recovery© is a patented and commercial process based on a gentle hydrolysis followed by spray defibration.

## 6. Biological Wood Decontamination

The cost and environmental impact of storage and incineration of wood waste such as wood impregnated with Cu-based preservative or fiber based panel board (MDF, HDF) are very high, and recycling or upcycling of these waste has great potential on global warming and carbon sequestration in wood industry [41,42]. Considerable efforts of research have been carried out to explore the bioremediation potential of wood decay bacteria and fungi in the contaminant’s removal from wood or water effluents [43,44,45,46,47]. Several studies have shown that fungi and bacteria are tolerant to different constituents of preservatives such as copper [48], CCA, and creosote [49,50]. The lactic acid bacteria *Lactobacillus bulgaricus* and *Streptococcus thermophilus* have been used to extract up to 93% copper, 86.5% chromium, and 97.8% of the arsenic after 8 days of fermentation at laboratory scale [51]. Fungi belonging to Basidiomycota have particularly interesting wood decontamination capacities [43]. Because of their ability to degrade the aromatic nuclei of lignin, the fungi that cause white rot are also able to assimilate the cyclic molecules contained in preservatives such as creosotes [45]. The use of basidiomycetes fungi causing brown rot for the extraction of heavy metals from wood has also shown good results, especially when this approach is coupled with pre-extraction with a weak acid such as citric acid [52,53]. The tolerance of brown rot fungi to copper fungi is due to their ability to excrete significant amounts of oxalic acid [54,55]. Oxalic acid is a strong organic acid and a good chelating agent capable of forming soluble Cr-oxalate $\mathrm{Cr}{\left(C_{2}O_{4}\right)}_{3}^{3-}$ and Cu-oxalate complexes $\mathrm{Cu}{\left(C_{2}O_{4}\right)}_{2}^{2-}$ and facilitates their extraction from wood [52,56,57]. However, all these results were carried out at the laboratory scale and the physiological mechanisms implemented by the microorganisms are still poorly known [43,58]. Despite several patents being published [59,60,61], no industrial process is available yet.

In order to provide new knowledge on the mechanisms involved in the biological decontamination of adjuvanted wood, the model fungus *Phanerochaete chrysosporium* was incubated in co-culture with a bacterial consortium with industrially treated wood with Cu/azole products. Promising results on a laboratory scale have been obtained (Figure 5).

The biological decontamination process seems to involve fungal–bacterial interaction for its completion. In order to improve the decontamination yields, the use of steam explosion as a pre-treatment phase is currently being tested. Once copper and other adjuvant have been removed, wood can be used for multiple material applications (panels, composite wood) or synthesis of molecules (production of synthons by biotransformation for polymers, energy, and agronomy.).

## 7. Spectroscopic Methods for Waste Wood Sorting

After collection, recoverable wood waste must be sorted to separate it from other types of waste such as plastics or metals. Density and flotation sorting methods separates materials according to their density, and metals are separated by magnetic methods. However, these classical methods have limitations and cannot discard MDF and heavily polluted wood. The wood treated with preservatives products containing heavy metals can be efficiently detected and sorted out, thanks to X-ray fluorescence spectroscopy (XRF) [62,63,64].

To recycle wood waste properly, waste furnishings items must be sorted to remove pollutants and contaminating materials. The most advanced technology for automated sorting is based on a spectroscopic technique using a near infrared (NIR) sensor coupled with blowing nozzles to separate elements according to their composition [65].

Wastes are subjected to radiations with wavelengths between 0.7 and 2.5 µm. Depending on their chemical composition, materials absorb some wavelengths, while others are captured and transmitted to a spectrometer, and these residual wavelengths are analyzed spectrally and spatially by a computer. The second derivatives of the spectra are calculated to perform a principal component analysis (PCA). This statistical analysis reduces the set of raw data to a limited set of significant and independent variables. The synthesized information is then represented as a scatter plot on a graph. One component is a combination of all the loadings of the different variables. To know the relationships between the variables, the principal components loadings are displayed on a plot. Based on those values, the meaning and contribution of each variable can be determined.

Near infrared spectroscopy can be used to optimize the removal of pollutants in recycled wood. Thanks to near infrared spectroscopy, the main components of wood were characterized. The signals of each characteristic functional group of wood components have been assigned by [66] and allow the identification of wood compounds in the near infrared.

NIR spectroscopy can be a solution to distinguish polluted wood from non-polluted wood. A principal component analysis was performed on four wood products with different composition. Figure 6A shows the result.

The principal component analysis yields four well-separated scatterplots, each representing a wood product. The principal component analysis separates the different materials: the wood waste, the MDF, and the glueless fibers. The treated MDF is also separated from the untreated MDF. The wood waste scatterplot is the largest due to its the variability. To explain the observed separations, the average NIR spectra of the wood products are shown in Figure 6B.

Each spectrum has a different profile. The greater the amplitude of a peak, the greater the difference between the samples. Each peak corresponds to a range of wavelengths that is attributed to a chemical bond. The differences between the peaks explain the differences in composition between the composites. NIR analysis can thus separate different wood products thanks to the differences in their compositions.

This analysis can go further. The work carried out by [65] has shown the potential of NIR to differentiate wood–plastic composites (WPC). However, this promising technique for the identification of certain plastics is limited for the detection of black plastics in recycled wood. In order to eliminate this gray area, authors of [67] studied the interest of mid-infrared for the detection of black plastics. In mid-infrared, black plastics have characteristic and identifiable spectra.

## 8. Microscopic Methods for Waste Wood Characterization

Developments in the fields of imaging and control of the conditions under which microorganisms are cultivated in fermenters open up the possibility of characterizing changes in the wood material and better deciphering the processes involved in recycling processes.

Indeed, when developing a recycling process, it is often essential to be able to observe changes in the material. In addition to conventional transmission photon microscopy, more advanced methods such as confocal laser scanning microscopy (CLSM) or scanning electron microscopy coupled to microanalysis (EDS/WDS-SEM) allow in situ investigation of wood waste during processing. As these techniques are widely used in the literature, we will give here some examples of results obtained during laboratory-scale processes.

The monitoring of copper detoxification by a filamentous fungus involves the sampling of wood particles that could be analyzed by different types of imaging techniques (Figure 7).

In bio-remediation processes of wood containing heavy metals such as copper, scanning electron microscopy (SEM) coupled with energy-dispersive X-ray spectroscopy (EDS) can provide valuable information on the relocation of contaminants mediated by the fungus. Industrial wood impregnated with commercial formulated preservative product was ground in particles of 2–5 mm. After treatment, wood particles containing 2700 ppm of copper (XRF quantification) were incubated with the fungus/bacteria consortium.

In the case of organic contaminants such as urea formaldehyde resin, fluorescence microscopy, and more specifically confocal laser scanning microscopy (CLSM), can be used to visualize the constituents of interest in situ if they are fluorescent or if they can be stained by fluorescent dyes [68,69,70]. Urea-formaldehyde (UF) resin removal from wood fibers in MDF panels waste after steam explosion can be visualized by CLSM using spectral deconvolution. This imaging technology is widely used in botanical sciences [71,72]. Once spectral calibration is achieved, online spectral unmixing (hyperspectral imaging) of the fluorescence emission signal becomes possible. Thus, the UF removal efficiency of a process can be evaluated (Figure 8). Microscopic inspection of the sample provides complementary information that might be hardly available by the use of analytic chemistry.

The use of microscopy in the monitoring and analysis of the mechanisms involved in a wood waste treatment process, in combination with other analytical methods, makes it possible to visualize and better understand the various transformations of the material as well as the physiological mechanisms developed by the microorganisms.

## 9. Conclusions and Recommendations

The current waste wood recycling processes need to be improved in order to (1) prioritize material recovery over energy recovery by cascading and (2) to limit as much as possible the non-recyclable batches, which may contain at least partly highly contaminated class C wood and/or MDF. From this study and on a European scale, the following conclusions and recommendations can be drawn:

First of all, the implementation of a more relevant harmonized European classification for a cascade utilization of waste wood is needed.

A better knowledge of wood waste composition and quality and an improvement of the current sorting techniques are essential. Automated sorting techniques based on spectrometric detectors, using, in particular, medium and near infrared radiation, could allow efficient separation according to the chemical composition of the waste.

An improvement in recycling routes can be achieved through a purification stage of the wood particles or fibers. This purification can be carried out by physicochemical methods. Biological approaches using fungi are also very promising.

The points discussed above required the development of the advanced techniques such as microscopy or diffraction techniques for a suitable characterization of contaminated wood.

Progress in all these fields will be required in the next few years to enable an effective cascading use of wood.

## Figures and Tables

**Figure 1 polymers-13-01752-f001:**
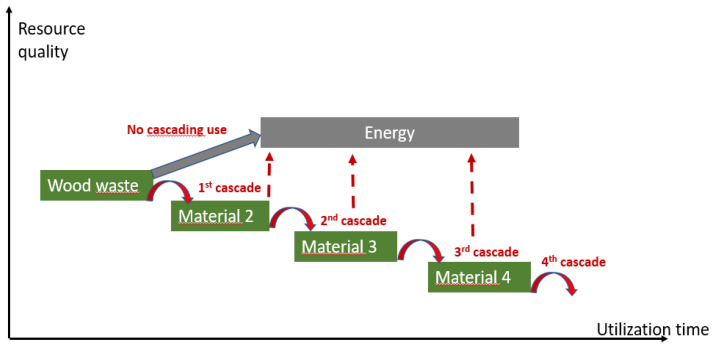
Cascading effect of wood waste.

**Figure 2 polymers-13-01752-f002:**
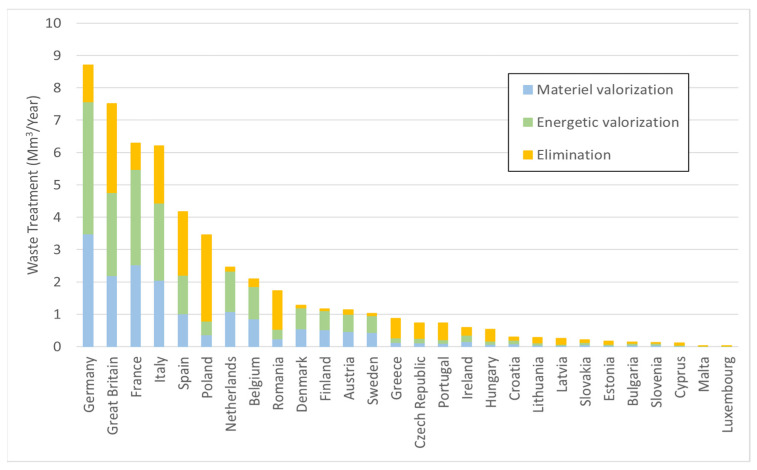
Wood waste volumes and handling methods in Europe. Drawn from [4].

**Figure 3 polymers-13-01752-f003:**
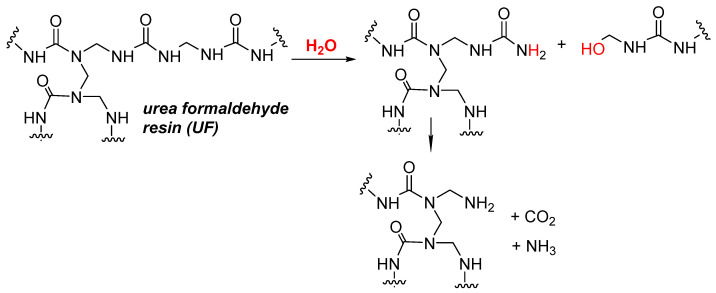
Chemical hydrolysis of urea—formaldehyde resin (adapted from [23]).

**Figure 4 polymers-13-01752-f004:**
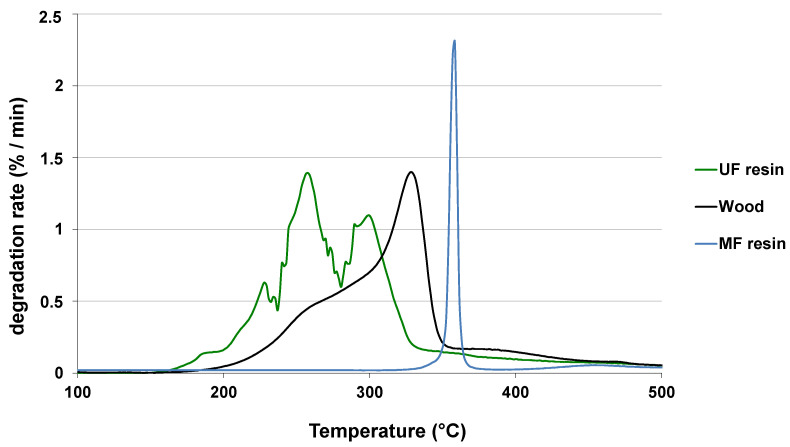
Degradation rate of wood, UF resin, and MF resin as a function of temperature [32].

**Figure 5 polymers-13-01752-f005:**
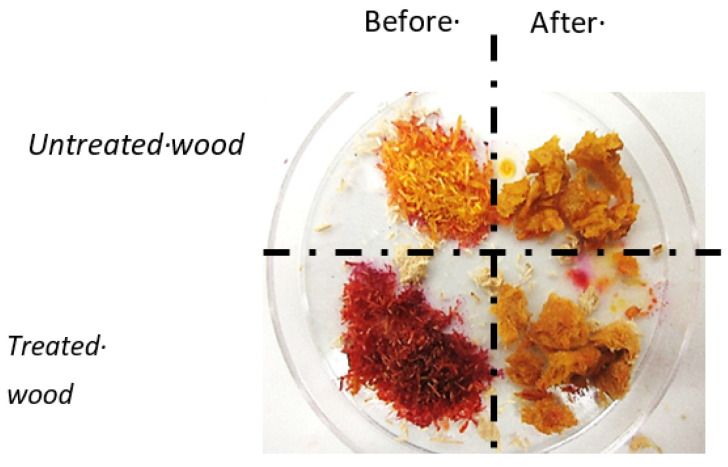
Colorimetric copper with (Pyridyl-2-azo)-1-naphthol-2 (PAN) in milled wood sample before or after bioprocessing treatment. The loss of wood stain shows the decontamination of copper from wood by microorganisms.

**Figure 6 polymers-13-01752-f006:**
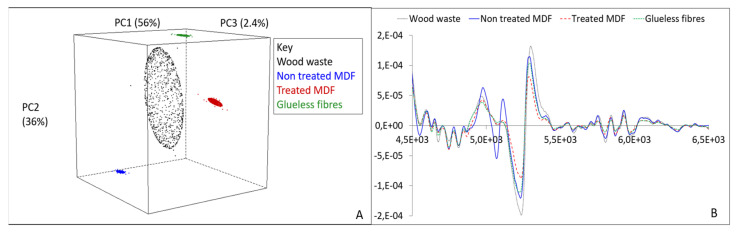
Principal component analysis performed on four wood products-plastic composites (**A**) and loadings analysis (**B**) (non-published data).

**Figure 7 polymers-13-01752-f007:**
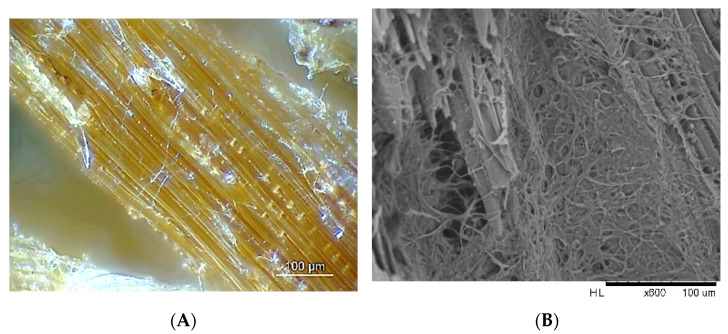
Microscopic monitoring of filamentous fungus growth during a bioremediation process: episcopy microscopy (**A**) and scanning electron microscopy (**B**), non-published data.

**Figure 8 polymers-13-01752-f008:**
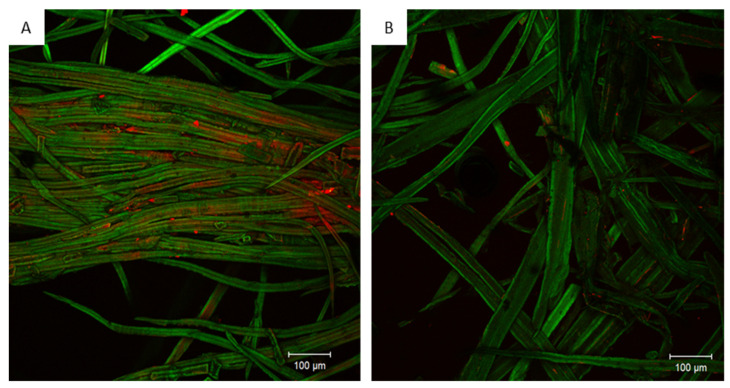
CLSM image showing the location of the resin (red) on wood fiber (green) before (**A**) and after (**B**) decontamination treatment, non-published data.

**Table 1 polymers-13-01752-t001:** Classification of wood waste according to its origin (adapted from [15]).

Origin	Type	Class
Packaging	Pallets and boxes (untreated, no MDF) Pallets and boxes (with MDF/treated wood)	1–2 3
Construction/demolition	Wood from construction and rebuilding (untreated, no MDF) Old wood from demolition and rebuilding (with MDF/treated wood)	1–2 3
Furnitures	Furniture (untreated, no fiberboard) Furniture (with fiberboard and/or treated wood) Furniture, upholstered	1–2 3 3
Others	Impregnated wood (wood treated with CCA, creosote or PCP) Composite building materials from demolition Miscellaneous (items made out of plastic, glass, metal, cardboard)	4 3 3

**Table 2 polymers-13-01752-t002:** Limit thresholds of contamination of recycled wood for panel production and of the panels themselves [17,18].

Elements	Limit Values (mg/Kg Wood)
Arsenic (As)	25
Cadmium (Cd)	50
Chromium (Cr)	25
Copper (Cu)	40
Lead (Pb)	90
Mercury (Hg)	25
Fluorine (F)	100
Chlorine (Cl)	1000
Pentachlorophenol (PCP)	5
Creosote (Benzo(a)pyrene)	0.5

**Table 3 polymers-13-01752-t003:** Classification under discussion at the French level for wood waste [22].

Class (UK/Fin— Ger)	Chemical Composition	Origin	Panels Recycling Use	Energy Production
1 (A—AI)	Clean biomass	Packaging, solid wood offcuts	Yes	Yes (2910-A)
2 (B—AII)	Organohalogen and heavy metal thresholds	Building, waste from furniture	Yes	Yes(2910-B)
3 (C—AIII)	Other wood without hazardous substances	Waste from furniture, mixed waste	Yes or no depending on composition	Yes (2971 or 2771 depending on type)
4 (D—AIV)	Hazardous wastes	Exterior fittings (creosote, CCA, Cu-azole)	No	Yes (2770)

## Data Availability

Not applicable.

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
