# Peer review of "Cascading Recycling of Wood Waste: A Review"

_polymers, 2021, doi:10.3390/polym13111752_

Round 1

Reviewer 1 Report

The manuscript is focused on an important and innovative research topic, namely recent developments and advances in cascading use and recycling of wood waste.

In lines 3-7, please carefully check the provided authors’ information and affiliations and revise it accordingly, e.g. the authors’ names are given twice (lines 3 and 5), and no information about the third affiliation is provided.

The abstract of the manuscript (lines 11 to 22) and the keywords (lines 24-25) correspond to the title, aims and objectives of the manuscript.

In the abstract, line 18, the authors stated this is a ‘chapter’. I’d recommend to change it to ‘review’, which is more appropriate.

In lines 37-38, there is the statement “....to meet global demand in Europe.” which does not sound very logical, it is either global (worldwide) or European demand, please revise.

In line 43, Figure 1, please provide photo credits, e.g. (photo by…).

In line 45, please delete the unnecessary “in itself”.

After the text in lines 45-53, I’d recommend to include the definition of cascading use of wood resources - “the efficient utilisation of resources by using residues and recycled materials for material use to extend total biomass availability within a given system”.

In addition, I’d also suggest to include information and data about the so-called post-consumer wood - wood products that are disposed at the end of their life cycle, e.g. wooden furniture, window frames and wood-based panels, doors, windows, various construction materials, etc., as well as relevant references.

Please provide some reference for Figure 2 in the text of the manuscript.

In lines 92-93, provide relevant reference for the statement.

In line 116, please provide the full terms of the abbreviations used in the text, i.e. polycyclic aromatic hydrocarbon (PAH), Pentachlorophenol (PCP), etc.

In line 129-130, the title of Table 1 should be given above the table, not below it, please revise.

In line 168, please use “particleboard” instead of “chipboard”.

In line 180, please revise the title of Figure 4 by providing the reference in square brackets.

In line 202, please provide the full term, i.e. melamine formaldehyde resin, followed by the abbreviation MF.

The information, provided in section 4, Wood fibres and MDF, is generally true and reflects the existing situation. However, the addition of more references is highly recommended. There are many successfully reported attempts to produce fibreboards from recycled fibres, please check the following examples:

https://doi.org/10.1080/17480272.2018.1528479

https://doi.org/10.5658/WOOD.2017.45.3.297

https://doi.org/10.3390/f11060613

https://doi.org/10.1007/s12649-021-01391-4

https://doi.org/10.1080/02773813.2017.1316741

https://doi.org/10.3390/polym13040639

https://doi.org/10.1007/s001070050268

https://doi.org/10.1007/s13196-017-0198-6

The authors stated that recycled wood is predominantly used in the production of particleboards (lines 15, 58, 112), but the information is a too general without any particular data presented and/or discussed.

In general, the Conclusions (lines 394-416) are consistent with the results and reflect the main findings of the study. However, they are too general and have to be revised.

The references cited are appropriate and correspond to the topic of the manuscript. The inclusion of additional references in all sections of the manuscript is highly recommended. This will significantly increase the value of the presented review article.

In addition, the text of the manuscript and most of the references are not formatted in accordance with the journal requirements, please check the instruction for authors.

Author Response

Thank you for the detailed correction and constructive remarks. The manuscript has been greatly modified and improved accordingly. The corrections are highlighted in yellow in the text. 

All the minor corrections have been done. 

The paper has been strengthened with additional discussions and references (+31 references)  and the conclusion was re writed. 

Nicolas Brosse

Reviewer 2 Report

This manuscript gives a review on wood waste flows and volumes available in Europe, the current French and European legislation and the innovations under development in this field: innovative automated sorting techniques, physical-chemical processes for cleaning residual glue from the surface of wood particles, cleaning of MDF and bioremediation processes for cleaning hazardous wood contaminated by heavy metals or creosote. This manuscript has practical significance. However, major revisions are requested. My comments are as below:

  1. Figure 1 is common, and can not be used in a research paper.
  2. Figure 2 is too simple to give more information. Please expand it and give more information.
  3. The data in Figure 3 is got in 2007, which is too old to give practical significance. Please update the latest results.
  4. Figure 4 is normal. It is well known for the readers. Based on this, it can be removed.
  5. Figure 7 is not specific. Please redraw it.
  6. Is the data in Figure 11 obtained by the authors? Or they just cite from some other publications. They should clear about this and give the source, since this is a review.
  7. For a topic review, 42 references are too few. Some content should be added. For example, are there any research on development of new functional materials using waste wood in Europe? Since there are a lot of such researches on this topic such as Composites Part B-Engineering, 2021, 214, 108744; JOURNAL OF ALLOYS AND COMPOUNDS, 2021, 858, 157706.

Author Response

The manuscript has been greatly modified and improved according to remarks. The corrections are highlighted in yellow in the text. 

Fig. 1 , 4 and 7 were removed. Fig 2 was not removed but we try to improve it. Data in Fig 3 were updated. 31 references were added. 

Nicolas Brosse

Round 2

Reviewer 1 Report

Dear authors, all my comments and remarks have been properly addressed. Thank you.